# Efficient Large Multi-modal Models
# via Visual Context Compression

**Jieneng Chen**[*], **Luoxin Ye**[*], **Ju He**[*], **Zhao-Yang Wang, Daniel Khashabi**[†], **Alan Yuille**[†]
Johns Hopkins University

## Abstract

While significant advancements have been made in compressed representations for text embeddings in large language models (LLMs), the compression of *visual* tokens in multi-modal LLMs (MLLMs) has remained a largely overlooked area. In this work, we present the study on the analysis of redundancy concerning visual tokens and efficient training within these models. Our initial experiments show that eliminating up to 70% of visual tokens at the testing stage by simply average pooling only leads to a minimal 3% reduction in visual question answering accuracy on the GQA benchmark, indicating significant redundancy in visual context. Addressing this, we introduce *Visual Context Compressor*, which reduces the number of visual tokens to enhance training and inference efficiency without sacrificing performance. To minimize information loss caused by the compression on visual tokens while maintaining training efficiency, we develop *LLaVolta* as a light and staged training scheme that incorporates stage-wise visual context compression to progressively compress the visual tokens from heavily to lightly compression during training, yielding no loss of information when testing. Extensive experiments demonstrate that our approach enhances the performance of MLLMs in both image-language and video-language understanding, while also significantly cutting training costs and improving inference efficiency.

| | | |
|---|---|---|
| ⊕ | **Website** | https://beckschen.github.io/llavolta.html |
| ⚙ | **Code** | https://github.com/Beckschen/LLaVolta |

## 1 Introduction

The advent of LLMs [33, 34, 44] has marked a new era in the field of artificial intelligence and natural language processing. LLMs can play a role as a universal interface for a general-purpose assistant, where various task instructions can be explicitly represented in language and guide the end-to-end trained neural assistant to solve a task of interest. For example, the recent success of ChatGPT [33] and GPT-4 [34] have demonstrated the power of aligned LLMs in following human instructions, and have stimulated tremendous interest in developing open-source LLMs [41, 43]. As the horizon of LLM applications broadens and the availability of open-source LLMs increases, the integration of multi-modality into these models presents a new frontier in expanding their capabilities. Multi-modal LLMs [1, 28, 40, 54] (MLLMs), which can process and understand not just text but also visual information, stand at the cutting edge of this evolution.

While MLLMs have made significant strides, a crucial aspect that remains relatively unexplored is the efficient representation and processing of visual information within these models. Substantial efforts [18, 35, 53] have been dedicated to optimizing the efficient representation of text tokens through various compression techniques [18, 35, 53], aimed at enhancing inference efficiency by

---

[*]Contributed equally.
[†]Advised equally.

38th Conference on Neural Information Processing Systems (NeurIPS 2024).

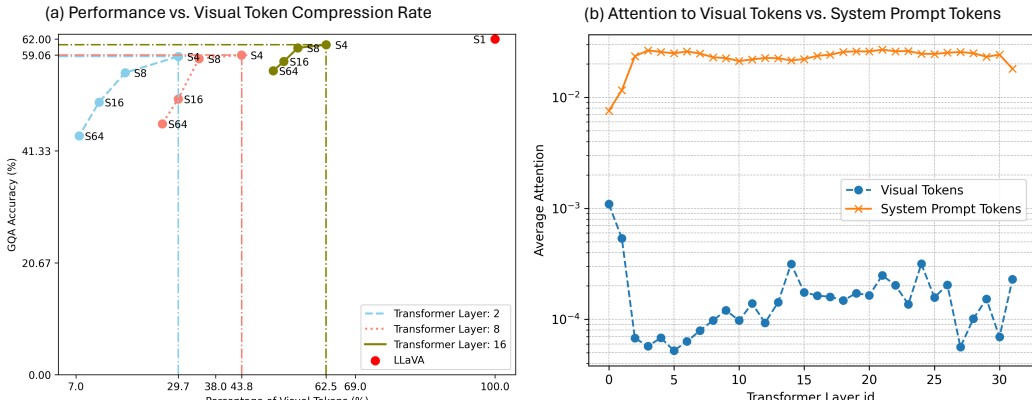

Figure 1: **Visual tokens are redundant in MLLMs. Left:** The accuracy of the LLaVA-1.5-7B [28] model(without re-train) on the GQA [20] benchmarks varies with different percentages of retained visual tokens. The $x$-axis represents the percentage of original visual tokens preserved after applying 1D average pooling with varying stride sizes $S$ applied in $i$-th Transformer layer. **Right:** Visual tokens receive less attention from the [ANS] token as we go deeper into its layers of LLaVA-1.5-7B model. These findings collectively suggest a significant redundancy within the visual tokens of the MLLMs.

attentively selecting important tokens. However, the efficient learning of *visual* tokens in MLLM has not garnered comparable attention. Naturally, this raises questions about the potential redundancy present in visual tokens and its implications for the overall computational efficiency of MLLMs.

We start our work by addressing the question: *Are visual tokens redundant in multi-modal LLMs?* To explore this, we first experiment with simply reducing the number of visual tokens in a pre-trained LLaVA-1.5-7B [28] at the inference stage via average pooling (§3.2). As shown in Fig.1 (left), our initial results demonstrate that eliminating up to 70% of visual tokens by pooling them with a stride of 4 starting from Transformer layer 2 incurs only a minimal performance loss on the GQA benchmark, specifically a 3% accuracy reduction. Additionally, we compute and present the average attention values from the [ANS] token to visual tokens and system prompt tokens across different Transformer layers in the pre-trained LLaVA-1.5-7B [28]. As revealed in Fig. 1 (right; blue trends), the visual tokens are generally less attended to, measured based on average attention from the [ANS] token, as the layers get deeper. These two early explorations indicate significant redundancy in visual tokens.

Addressing this, in this work we develop an effective *Visual Context Compressor* that can be integrated into the training of MLLMs. Surprisingly, a simple average pooler nested in LLMs stands out as the most effective compressor, outperforming the attention-based [18, 53] and parametric [23] counterparts. We attribute this to two reasons: (1) The simple pooling operation makes training stable, whereas prior attention-based approaches [18, 53] are specifically designed for accelerating inference rather than training. (2) Visual tokens in the deeper Transformer layers are less attended to (see Fig. 1 (right)) and particularly redundant, making a simple compressor placed in a deeper Transformer layer effective enough. At a lower training cost, the LLaVA-1.5-7B [28] trained with the proposed *Visual Context Compressor* is competitive with the non-compressed baseline across various multi-modal benchmarks (*e.g.*, GQA [20] and MM-Vet [50]). This dual achievement highlights *Visual Context Compressor*'s role as a pivotal advancement in enhancing the efficiency and performance of MLLMs across various multi-modal question-answering benchmarks.

To further mitigate the information loss caused by compressing visual tokens, especially under a large compression ratio (CR), we have devised a **LLaV**A-**p**owered **l**ite **tra**ining scheme, dubbed *LLaVolta*, which progressively employs *Visual Context Compressor* at multiple training stages with different compression ratios (§3.3). Specifically, *LLaVolta* progresses through several stages, beginning with a high level of visual token compression and gradually reducing the compression ratio until the final stages, where full visual tokens are utilized. This multi-stage approach allows for adaptive compression levels that ensure training efficiency without losing information at testing, thus maintaining the overall effectiveness of the model.

Extensive experimental evaluations of *LLaVolta* have been conducted on thirteen widely-adopted MLLM benchmarks for both image-language understanding and video-language understanding ,

showing promising results. We observe that *LLaVolta* not only enhances the performance of MLLMs, but also achieves a substantial reduction in training costs. These experiments validate the effectiveness of our method, demonstrating its capability to optimize resource utilization while maintaining or even improving model performance.

In summary, our paper makes the following contributions:

- We present two initial studies to verify the redundancy of visual tokens in MLLMs.

- We propose the *Visual Context Compressor*, a simple yet effective compression technique that utilizes an average pooler, enhancing the efficiency of multi-modal models.

- We propose the *LLaVolta* as an efficient training scheme by leveraging *Visual Context Compressor* at multiple training stages with a progressively decreasing compression ratio. To the best of our knowledge, we are among the first to explore efficient training of MLLMs.

- Extensive experiments show that our approach not only improves the performance of MLLMs in image-language and video-language understanding across various benchmarks but also showcases efficiency gains by reducing training costs by 16% and inference latency by 24%.

## 2  Related Works

**Multi-modal LLMs.** The evolution of large language models [10, 33, 34] into their multi-modal counterparts [28, 40] represents a significant leap in their ability to follow instructions and generalize across tasks. This transition has been marked by seminal works such as Flamingo [1], BLIP-2 [23] and LLaVA [28], which have extended LLM capabilities to encompass visual tasks, demonstrating impressive zero-shot generalization and in-context learning abilities. Progress in multi-modal LLMs has primarily been driven by advancements in visual instruction tuning [28, 54], leveraging vision-language datasets and refining visual instruction-following data. Additionally, efforts have been made to enhance the grounding capabilities of multi-modal LLMs through the use of specialized datasets aimed at improving task-specific performance. Despite these advancements, the exploration of visual compression within multi-modal LLMs remains relatively underdeveloped. The design and optimization of compression strategies are crucial for maximizing the effectiveness and efficiency of multi-modal LLMs, suggesting a potential area for future research and development.

**Visual Redundancy.** In computer vision, reducing redundancy is crucial for creating efficient yet effective models without losing accuracy [4]. Redundancy in images often arises from the inherent characteristics of natural scenes, including repetitive patterns, textures, and areas of uniform color. These features, while contributing to the richness and detail of visual perception, can lead to inefficiencies in both storage and processing when not adequately addressed. Image compression algorithms [46] can reduce file size by eliminating or efficiently encoding redundant data. These methods take advantage of human visual perception's tolerances to subtly reduce data without significantly impacting image quality. Advanced machine learning models, particularly CNNs and autoencoders [3], offer sophisticated approaches to minimizing redundancy. Transformers [45], as a fundamental architecture for LLMs [10, 34], apply self-attention mechanisms to dynamically bind the most informative parts of tokens. Vision Transformers [6, 7, 8, 12, 16] trained with CLIP objective [7, 36] encode an image to a sequence of visual features for multi-modal LLMs [28]. Nevertheless, visual tokens receive less attention in LLMs due to attention shrinkage [47], resulting a waste of computation. In this work, we focus on reducing the redundancy of visual tokens in MLLMs.

**Efficient LLMs.** Efficient inference and training for LLMs are important. Compressing input sequences for efficiency reasons in Transformers is not a new idea for NLP. Much work is being done to accelerate the inference of LMs. For example, Pyramid Transformer variants [11] and [19] are proposed in Encoder-Decoder LMs that progressively compress the sequence as the layers grow deeper via pooling or core-set selection. Nawrot et al. [32] propose adaptively compressing the sequence based on the predicted semantic boundaries within the sequence. Rae et al. [37] propose compressing the fine-grained past activations to coarser memories. VCC [53] compress the sequence into a much smaller representation at each layer by prioritizing important tokens. Besides efficient inference, accelerating training for LLMs attracts attention as well. A staged training setup [38] is proposed which begins with a small model and incrementally increases the amount of compute used

for training by applying a growth operator to increase the model depth and width. However, efficient training for LLMs in multi-modal scenarios is rarely explored.

## 3 Method

In this section, we first introduce an overview of multi-modal LLMs in § 3.1. Then, we define the problem of visual redundancy and introduce *Visual Context Compressor* in § 3.2. Finally, we present our proposed *LLaVolta* in § 3.3.

### 3.1 Preliminaries: A Multi-modal LLM

We start by reviewing the design of the LLaVA family [27, 28]. For processing an input image $\mathbf{X}_v$, we utilize the pre-trained CLIP visual encoder ViT-L/14, as detailed by [36], to extract the visual feature $\mathbf{Z}_v = g(\mathbf{X}_v)$, where $g(.)$ indicates the visual encoder. To bridge the gap between visual and linguistic modalities, the LLaVA [27, 28] framework as an MLLM implements a straightforward linear/MLP transformation. This involves a trainable projection matrix $\mathbf{W}$, which maps the visual features $\mathbf{Z}_v$ into the linguistic embedding space, producing language embedding tokens $\mathbf{H}_v = \mathbf{W}\mathbf{Z}_v$. These tokens are designed to match the dimensionality of the word embeddings within the LLM.

For each image $\mathbf{X}_v$, one can generate multi-turn conversation data $(\mathbf{X}_q^1, \mathbf{X}_a^1, \cdots, \mathbf{X}_q^T, \mathbf{X}_a^T)$ with $T$ as the number of turns. One can organize them as a sequence, by treating all answers as the assistant's response and the instruction $\mathbf{X}_{\texttt{instruct}}^t$ at the $t$-th turn as:

$$\mathbf{X}_{\texttt{instruct}}^t = \left\{ \begin{array}{ll} \text{Random Choose}[\mathbf{X}_q^1, \mathbf{X}_v] \text{ or } [\mathbf{X}_v, \mathbf{X}_q^1], & t = 1 \\ \mathbf{X}_q^t, & t > 1 \end{array} \right. \tag{1}$$

This approach establishes a standardized format for the multi-modal instruction-following sequence. It allows for the instruction-based tuning of the LLM to be applied to the prediction tokens, utilizing the model's native auto-regressive training objective. Specifically, for a sequence with length $L$, the likelihood of the target responses $\mathbf{X}_a$ is calculated as:

$$p(\mathbf{X}_a | \mathbf{X}_v, \mathbf{X}_{\texttt{instruct}}) = \prod_{i=1}^{L} p_\theta(x_i | \mathbf{X}_v, \mathbf{X}_{\texttt{instruct}, <i}, \mathbf{X}_{a, <i}), \tag{2}$$

### 3.2 *Visual Context Compressor*

**Problem Formulation**: The redundancy observed in images often arises from inherent traits of natural scenes, including repetitive patterns, textures, and regions with uniform color. While these traits enrich visual perception by offering detail and depth, they can also present challenges in terms of storage and processing efficiency. Considering the inherent limitations of Transformers in handling long sequences [2, 29, 49], it is critical to minimize any length redundancies to obtain a more effective accuracy/efficiency trade-off.

The objective of this study is to decrease the length of visual tokens $\mathbf{X}_v$ (*i.e.,* its hidden states $\mathbf{H}_v$ if inside LLMs), while simultaneously maximizing the probability of the target response $p(\mathbf{X}_a | \mathbf{X}_v, \mathbf{X}_{\texttt{instruct}})$ as described in Equation (2).

*Visual Context Compressor*: A key design change that we introduce is a compressor layer that compresses the dimensions of the visual inputs by reducing the effective number of visual tokens. As depicted in Fig. 2, the compressor is simply an average pooler in our setting. It is applied to the visual tokens in $k$-th Transformer layer of an LLM. Formally, given the hidden visual tokens at $k$-th Transformer layer $\mathbf{H}_k \in \mathbb{R}^{B \times C \times L}$, the compressor is expected to fulfill the following projection: $f : \mathbb{R}^{B \times C \times L} \mapsto \mathbb{R}^{B \times C \times L_{out}}$, which results in compressed visual tokens $\tilde{\mathbf{H}}_k \in \mathbb{R}^{B \times C \times L_{out}}$, where $L_{out} = \frac{L}{S}$ with $s$ as the compression stride. In §4, we explore multiple variants of compressor $f$ to reduce the token length, including random token dropping [17] with dropping ratio $1 - \frac{1}{S}$, K-Means [21] with number of centroids set to $N_C = \frac{L}{S}$, attention-based token-centric compression [53], attention-based token dropping [9, 18], and average pooling with stride $s$. To our surprise, we find that the simple average pooler is the most effective compressor for vision tokens within MLLMs, due to its stability during training detailed in § 4.4. Thus, we choose average pooler as the compressor.

Note that the proposed *Visual Context Compressor* can be directly applied to any off-the-shelf MLLMs to assess the visual redundancy, as conducted in §4.2. One can also train an MLLM with *Visual Context Compressor* to reduce the number of visual tokens while maintaining competitive multi-modal performance.

**Compression Ratio (CR)**[3]. For an LLM with $N$ Transformer decoder layers, the compression ratio for visual tokens can be calculated as:

$$\text{CR} = \frac{N \cdot L}{(N - K) \cdot L_{out} + K \cdot L} \quad , \quad (3)$$

where $K$ is the $K$-th Transformer layer of a multi-modal LLM; $L$ is the the length of visual tokens input into Visual Context Compressor; $L_{out}$ is the compressed length of visual tokens generated by Visual Context Compressor, as illustrated in Fig. 2.

Our architecture modifications thus far mostly impacts the inference efficiency of MLLM, however, its impact on performance-compression trade-off remains unclear. We will study this question in the context of **training** MLLMs with a goal of enhancing efficiency without compromising performance. We then move on to further utilize *Visual Context Compressor* to design an efficient training scheme to incorporates Visual Context Compressor at various stages of the training process.

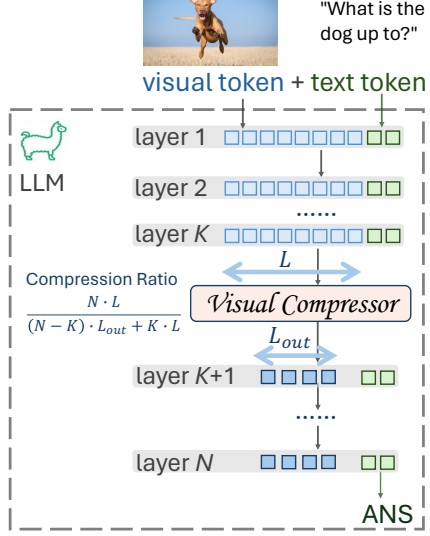

Figure 2: Example of Visual Context Compressor in a multi-modal LLM.

### 3.3 *LLaVolta* as a Light, Staged Training Scheme

Training with *Visual Context Compressor* not only facilitates efficient inference but also enhances training efficiency. However, devising an effective training scheme poses challenges when ensuring fair comparisons with the original LLaVA [27], primarily due to differences in the number of tokens involved in inference. This discrepancy may lead to information loss, particularly when operating under a scenario with a high compression ratio. To tackle this issue, we have developed a lite training scheme for LLaVA, dubbed as *LLaVolta*, which employs stage-wise visual context compression. Generally, assuming there are $N_s$ total stages, stage $i$ involves $\frac{1}{N_s}$ of the total training epochs with a compression ratio of $r_i$, and the final stage proceeds without any compression. Essentially, as training progresses, $i$ increases while $r_i$ decreases.

In this work, as depicted in Fig. 3, we primarily explore a three-stage training pipeline that progressively reduces the compression ratio, as detailed below:

**Training Stage I: Heavy Compression**. The MLLM training at the first one-third of the total training iterations commences with a heavy compression ratio ($> 500\%$), where *Visual Context Compressor* is applied in an early layer of the LLM with a large pooling stride. This setup enables a very fast training speed.

**Training Stage II: Light Compression**. The MLLM continues training with another one-third of the total training epochs. At this stage, *Visual Context Compressor* is applied at only the deeper layers of the LLM with a smaller pooling stride compared to Training Stage I.

**Training Stage III: No/subtle Compression**. The MLLM continues training during the final one-third of the total epochs, with either no compression or subtle compression applied. This stage is designed to align with the inference process, where visual tokens may also undergo compression. By maintaining consistency between training and inference, this approach ensures that critical information is preserved while still allowing for compression, minimizing any potential discrepancies between training and real-world use.

Given the above meta framework, we can instantiate a family of training schemes, as demonstrated in Tab. 1. The single-stage (non-compression) scheme is equivalent to the MLLM baseline. For

---

[3]Definition of compression ratio from Wikipedia

multi-stage training, the compression stage can either go deeper or wider. "deeper" implies an increase in $K$ (Transformer layer), while "wider" means a decrease in the stride of the pooler.

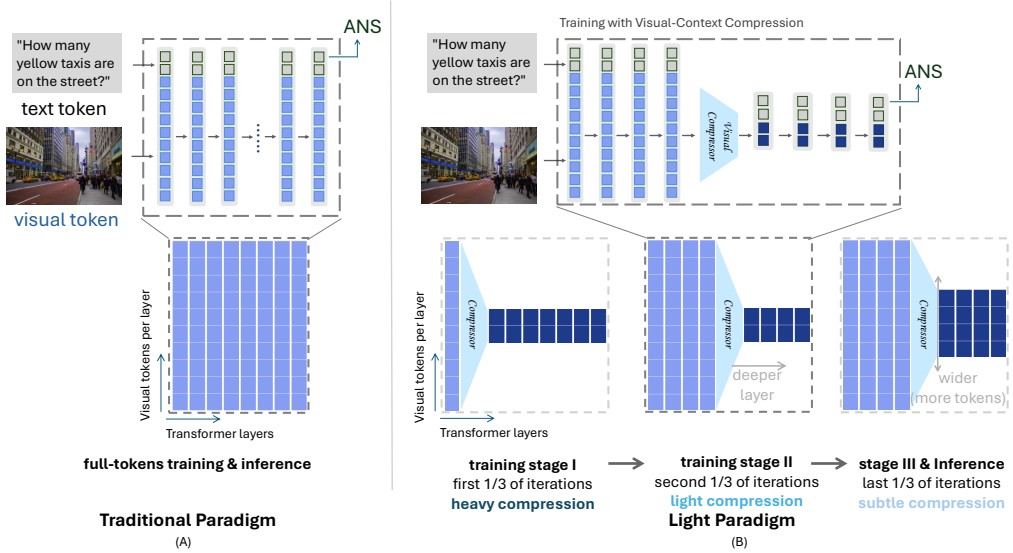

Figure 3: Training & inference paradigm comparison for conventional setting (A) and *LLaVolta* (B). Meta framework of *LLaVolta* consists three training stages: Stage I with heavy visual compression; Stage II with light visual compression in *deeper* layer; Stage III with subtle compression with *wider* token window without loss of performance. This can accelerate the training and inference by 18+% while maintaining performance.

| #Stages | Scheme | Stage | Layer | Stride | CR | #Epoch |
|---|---|---|---|---|---|---|
| Single | no compression | $S1$ | / | / | 100% | 1 |
| Two | compression | $S1$ | 2 | 8 | 557% | 0.5 |
| | | $S2$ | / | / | 100% | 0.5 |
| Three | compr. deeper | $S1$ | 2 | 8 | 557% | 0.33 |
| | | $S2$ | 16 | 8 | 178% | 0.33 |
| | | $S3$ | / | / | 100% | 0.33 |
| Three | compr. wider | $S1$ | 2 | 8 | 557% | 0.33 |
| | | $S2$ | 2 | 2 | 188% | 0.33 |
| | | $S3$ | / | / | 100% | 0.33 |

| #Stages | Scheme | Stage | Layer | Stride | CR | #Epoch |
|---|---|---|---|---|---|---|
| Four | wider then deeper | $S1$ | 2 | 8 | 557% | 0.25 |
| | | $S2$ | 2 | 2 | 188% | 0.25 |
| | | $S3$ | 16 | 2 | 133% | 0.25 |
| | | $S4$ | / | / | 100% | 0.25 |
| Four | deeper then wider | $S1$ | 2 | 8 | 557% | 0.25 |
| | | $S2$ | 16 | 8 | 178% | 0.25 |
| | | $S3$ | 16 | 2 | 133% | 0.25 |
| | | $S4$ | / | / | 100% | 0.25 |
| Three | last stage compression | $S1$ | 2 | 16 | 825% | 0.33 |
| | | $S2$ | 16 | 16 | 188% | 0.33 |
| | | $S3$ | 16 | 4 | 160% | 0.33 |

Table 1: **Instantiations of *LLaVolta* schemes**. deeper indicates that the compressor's position in the LLM shifts from the shallow layer (*e.g.*, 2) to a deeper layer (*e.g.*, 16). wider indicates that the compressor's stride decreases while the number of visual tokens increases. Last stage compression refers to using compressor at last stage for efficient inference.

Note that all training schemes will be standardized to complete just one epoch. Thus, in the three-stage training, each stage will receive one third of an epoch, while in the four-stage training, each stage will receive one fourth of an epoch. Effects of non-uniform stage splitting are presented in the Appendix.

## 4    Experiments

In this section, we begin by detailing the experimental setup in § 4.1. Next, we elaborate on the proof-of-concept in Section § 4.2. Following this, we validate the proposed *LLaVolta* in § 4.3 with an ablation study in § 4.4. Finally, we assess the extensibility to video-language in § 4.5.

## 4.1 Experimental Setup

We adopt the Vicuna-v1.5-7B [10] as the language model, leveraging the LLaMA2 codebase [43]. We leverage the pre-trained CLIP ViT-L/14 [12, 36] with an input resolution of $336 \times 336$, resulting in 576 visual tokens. We employ the LLaVA framework [27] to connect the frozen CLIP vision encoder and the Vicuna LLMs. Along with the projector, we train the entire LLM instead of parameter-efficient finetuning. We follow LLaVA-1.5 [27] to perform data preparation and training schedule for pretraining and instruction tuning. We conduct all the experiments with the machine of $8 \times$ Nvidia RTX 6000 Ada. Due to multiple invalid image links in the dataset of instruction tuning stage, the scores of LLaVA-1.5 reported in our analysis are reproduced by ourselves to ensure a fair comparison under the same experimental environment.

It is worth mentioning that assessing visual token redundancy only necessitates the inference of existing off-the-shelf models, whereas the other experiments involve the training of multi-modal LLMs, specifically projectors and LLMs.

**Benchmarks and Metrics**: We adopt thirteen benchmarks specifically designed for MLLM evaluation, including GQA [20], MM-Vet [50], ScienceQA (SQA)[31], MME[13], TextVQA [39], POPE [24], MMBench [30], MMBench-CN [30], VQA-v2 [14], LLaVA-Bench-in-the-Wild (LLaVA$^W$) [28], VisWiz [15], SEED-Image [22] and MMMU [52]. GQA and VQA-v2 evaluate the model's visual perception capabilities on open-ended short answers. MME-Perception evaluates model's visual perception with yes/no questions. ScienceQA with multiple choice are used to evaluate the zero-shot generalization on scientific question answering. TextVQA contains text-rich visual question answering. MMBench and the CN version evaluate a model's answer robustness with all-round shuffling on multiple choice answers. MM-Vet evaluates a model's capabilities in engaging in visual conversations. Additionally, we extend *LLaVolta* to video-language understanding, and follow Video-LLaVA [26] to evaluate the models on MSVD-QA [5], MSRVTT-QA [48] and ActivityNet-QA [51], where the accuracy and score are assessed using GPT-Assistant.
We report the official metrics calculated using the standard implementations provided for each benchmark for a fair comparison. Latency is reported as the time taken during inference until the first answer token is produced. When reporting average performance in Table 2, the score of MME is divided by 2000, as its range is from 800 to 2000. TFLOPs are profiled via DeepSpeed. For total number of tokens, $\#\text{Tokens} = \sum_i^N \#\text{Token}^i$. The training time is reported for one epoch of training during the LLaVA instruction-tuning stage. The Compression Ratio (CR) is defined as in Equation 3.

## 4.2 Proof of Concept: Visual Context Redundancy

To assess the redundancy of visual tokens, we perform average pooling within an off-the-shelf LLaVA-1.5-7B checkpoint at the testing stage, using different pooling stride sizes $S$ across various Transformer layers $K$. As shown in Fig. 1, the model still exhibits strong performance even when retaining only 62.5% of the visual tokens ($S = 4, K = 16$) in the MM-Vet benchmark, without the need for additional training. When adopting the same setting ($S = 4, K = 16$), a similar trend can be observed in the GQA benchmark as well, where the compressed model only has 1% performance drop than the uncompressed counterpart. Surprisingly, in the GQA benchmark, eliminating up to 70% of visual tokens ($S = 4, K = 16$) results in a mere 3% decrease in performance. This proof-of-concept shows a certain level of redundancy in the visual tokens within MLLMs.

## 4.3 Main Results: *LLaVolta*

In this section, we present the main results of *LLaVolta* schemes instantiated in § 3.3. We conduct a thorough evaluation of the multi-modal capability across 13 benchmarks. Tab. 2 demonstrates that our proposed *LLaVolta* not only consistently lowers training costs by 19% (15.3 hours *vs*. 12.4 hours) but also surpasses the non-compression baseline. The last-stage-compression training schemes achieves the best performance across thirteen benchmarks and obtains 62.1% average performance, improving LLaVA-v1.5-7B [27] with much less inference TFLOPs and training time. This indicates the necessity of designing an optimally lite training scheme.

| #Stages | Scheme | #Tokens† | CR† | Last Stage TFLOPs† | Latency (ms) | Train Time | GQA | MMVet | SQA | MME | VQA$^T$ | POPE | MMB | MMB$^{CN}$ | VQA$^{v2}$ | LLaVA$^w$ | VisWiz | SEED$^I$ | MMMU | Avg. |
|---|---|---|---|---|---|---|---|---|---|---|---|---|---|---|---|---|---|---|---|---|
| Single | no compression | 18432 | - | 8.26 | 68.5 | 15.3h | **62.6**$_{49}$ | **31.9**$_1$ | 70.8$_{59}$ | 1467$_{13}$ | 58.3$_{15}$ | 86.1$_{24}$ | 65.3$_{93}$ | 59.4$_{92}$ | **78.9**$_{37}$ | 65.5$_{56}$ | 49.8$_6$ | **66.7**$_{25}$ | 35.1$_{86}$ | 61.8$_{32}$ |
| Two | compression | 10062 | 183% | 8.26 | 68.5 | 12.8h | 61.9$_{23}$ | 31.7$_{1.5}$ | 70.9$_{34}$ | **1480**$_{23}$ | 58.3$_{46}$ | 86.5$_{33}$ | 64.8$_{23}$ | 59.0$_{1.1}$ | 78.5$_{20}$ | 67.3$_{91}$ | 47.2$_{1.8}$ | 64.9$_{17}$ | 34.9$_{11}$ | 61.5$_{40}$ |
| Three | compr. deeper | 10597 | 174% | 8.26 | 68.5 | 12.8h | 62.1$_{01}$ | 30.5$_{40}$ | 70.5$_{23}$ | 1477$_{13}$ | 58.4$_{07}$ | 86.6$_{14}$ | 65.6$_{26}$ | 59.9$_{27}$ | 78.5$_{22}$ | 67.5$_{1.4}$ | 49.2$_{56}$ | 65.9$_{17}$ | 35.0$_{19}$ | 61.8$_{10}$ |
| Three | compr. wider | 10407 | 177% | 8.26 | 68.5 | 12.8h | 61.1$_{1.6}$ | 31.8$_{61}$ | 71.0$_{28}$ | 1434$_{12}$ | 58.5$_{04}$ | 86.6$_{06}$ | 64.8$_{23}$ | 59.1$_{83}$ | 78.7$_{02}$ | 64.3$_{48}$ | 49.8$_{1.1}$ | 65.3$_{04}$ | 34.3$_{75}$ | 61.3$_{28}$ |
| Four | wider then deeper | 11088 | 166% | 8.26 | 68.5 | 12.9h | 62.1$_{09}$ | 31.6$_{58}$ | **71.4**$_{36}$ | 1444$_{15}$ | 58.7$_{24}$ | **86.8**$_{21}$ | 65.3$_{30}$ | 59.3$_{26}$ | 78.8$_{05}$ | 67.3$_{1.1}$ | **50.1**$_{21}$ | 65.6$_{15}$ | 33.8$_{78}$ | 61.8$_{35}$ |
| Four | deeper then wider | 10863 | 170% | 8.26 | 68.5 | 12.8h | 62.1$_{07}$ | 31.5$_{20}$ | 70.5$_{16}$ | 1472$_{16}$ | **58.7**$_{08}$ | 86.3$_{33}$ | **65.6**$_{52}$ | **59.9**$_{61}$ | 78.8$_{03}$ | 68.2$_{2.1}$ | 48.3$_{1.3}$ | 66.1$_{20}$ | 35.1$_{02}$ | 61.9$_{47}$ |
| Three | last stage compression | 7848 | 235% | 5.47 | 52.2 | 12.4h | 62.3$_{26}$ | 31.5$_{35}$ | 71.0$_{17}$ | **1519**$_{14}$ | 58.0$_{12}$ | 86.5$_{30}$ | 65.3$_{45}$ | 59.1$_{60}$ | 78.2$_{02}$ | **69.2**$_{0.15}$ | **50.2**$_{1.0}$ | 65.4$_{16}$ | **35.4**$_{22}$ | **62.1**$_{07}$ |

Table 2: **Performance of *LLaVolta***. See the definition of each training scheme in Tab. 1. †: average across stages. First five derived schemes for training acceleration achieve competitive results while reducing 16% training time. The last scheme, last stage compression, achieved the shortest training time (12.4 hours) and the lowest inference cost (5.47 TFLOPs), but also the highest average performance (62.1%). We report average results across three runs, with the standard deviation written at the bottom right of the average result. *The last stage compression training achieves the best average performance across thirteen benchmarks, outperforming the baseline (LLaVA-v1.5-7B) while requiring significantly less training time.*

## 4.4 Ablation Study

In this section, we perform an ablation study on the choice of visual compressors by comparing different compression methods. Additionally, we examine the effects of varying the stride and LLM layer in training *Visual Context Compressor*.

| Compressor | #Tokens | CR | GQA | MM-Vet | SQA | MME | VQA$^T$ | POPE | MMB | MMB$^{CN}$ | VQA$^{v2}$ | LLaVA$^w$ | VisWiz | SEED$^I$ | MMMU | Avg. |
|---|---|---|---|---|---|---|---|---|---|---|---|---|---|---|---|---|
| *Train **without** compression; Testing with compression* | | | | | | | | | | | | | | | | |
| Random Dropping | 3312 | 556% | 50.6 | 21.4 | 69.3 | 1142 | 46.5 | 55.8 | 39.7 | 33.3 | 59.3 | 47.6 | 47.2 | 52.2 | 34.3 | 47.3 |
| K-Means | 3312 | 556% | _54.4_ | 25.9 | **69.7** | 1155 | 49.0 | **78.6** | 55.3 | 46.1 | **69.3** | **57.6** | 48.9 | 56.1 | 32.9 | 54.0 |
| FastV [9] | 3312 | 556% | 52.1 | **30.6** | _69.4_ | **1298** | **53.4** | 65.6 | _60.1_ | **53.0** | _68.6_ | 54.8 | **50.0** | _56.3_ | **34.9** | **54.9** |
| VCC [53] | 3582 | 514% | **54.7** | _26.9_ | 69.2 | _1246_ | _49.2_ | _72.3_ | 60.8 | _52.0_ | 68.1 | 55.6 | 47.8 | **57.0** | _34.8_ | _54.7_ |
| Average Pooling | 3312 | 556% | 53.7 | 25.6 | _69.4_ | 1150 | 47.7 | 70.1 | 56.4 | 46.5 | 67.0 | _55.6_ | _50.0_ | 55.7 | 34.3 | 53.0 |
| *Train **with** compression; Testing with compression* | | | | | | | | | | | | | | | | |
| Random Dropping | 3312 | 556% | 53.4 | 25.0 | 69.4 | 1186 | 49.4 | 64.9 | 52.0 | 41.1 | 59.7 | 51.5 | 47.9 | 52.6 | 34.6 | 50.8 |
| K-Means | 3312 | 556% | 57.5 | 25.9 | 55.6 | 1279 | 51.4 | 79.4 | 62.6 | 54.6 | _75.7_ | 59 | 46.1 | 59.2 | 34.1 | 57.9 |
| FastV [9] | 3312 | 556% | 55.9 | 27.9 | 70.4 | 1327 | 49.7 | 79.8 | 62.9 | _55.9_ | 69.5 | _61.7_ | **49.6** | 56.8 | **35.1** | 57.0 |
| VCC [53] | 3582 | 514% | _57.7_ | _29.3_ | _70.7_ | _1398_ | _53.0_ | _83.6_ | **65.0** | 55.8 | 74.1 | 58.0 | _48.2_ | _60.1_ | _35.0_ | _58.5_ |
| Average Pooling | 3312 | 556% | **60.0** | **30.7** | **70.8** | **1450** | **55.1** | **85.5** | **65.0** | **59.5** | **75.9** | **66.9** | 46.4 | **62.6** | 33.8 | **60.4** |

Table 3: **Comparison among different visual compressors**. Higher values are preferred. All methods except VCC are set to the compression ratio of 556% to approximate VCC's 514% [53] for a fair comparison. The best scores are marked as gray and the second best are underlined. Attention-based compressors (*i.e.*, FastV and VCC) excel during the inference phase, yet their application to the training phase proves challenging. *Average pooling shows a more stable performance during the training phase.*

**Choice of Visual Compressors**. The design choices include (1) random token dropping, (2) K-Means clustering, (3) average pooling, (4) FastV [9], (5) VCC [18], (6) parametric pre-trained Q-Former [23]. We have the following three observations. Firstly, Tab. 3 shows that the attention-based methods, including FastV and VCC win 9/13 best and second best scores, showcasing the high performance when compressing visual tokens in inference. However, they are ineffective when applied to training because the in-training attention scores are unstable. Secondly, and surprisingly, the average pooling obtains the highest scores on eleven out of thirteen benchmarks when it is used to train MLLMs with a high CR. Thirdly, Tab. 4 shows that both Q-Former and average pooling can obtain reasonably good performance when trained with extremely high CRs, and the average pooling performs better with less training cost. The reason could be that the Q-Former resamples tokens outside the LLM, potentially causing the LLM to overlook crucial information relevant to the response. In contrast, our approach employs average pooling subsequent to Transformer layer $K$, allowing the initial $K$ layers of the LLM to effectively retain important information from uncompressed tokens. Given these three insights, we select average pooling as our favored approach for visual compression.

**Performance Across Compression Ratios**. Herein, we train the multi-modal LLM with our *Visual Context Compressor* in various settings. As demonstrated in Tab. 5, the proposed method offers certain improvements and trade-offs compared to the state-of-the-art method, LLaVA-1.5-7B. We have the following two observations. Firstly, in the heavy compression level, the performance of MLLM is inversely proportional to the compression ratio (linearly scaling to the number of visual

| Method | #Param | #Tokens | CR | Train Time | GQA | MMVet | SQA | MME | VQA$^T$ | POPE | MMB | MMB$^{CN}$ | VQA$^{v2}$ | LLaVA$^w$ | VisWiz | SEED$^I$ | MMMU | Avg. |
|---|---|---|---|---|---|---|---|---|---|---|---|---|---|---|---|---|---|---|
| Q-Former [23] | 105M | 1024 | 1800% | 10.4h | 55.7 | 26.4 | 69.3 | 1217 | 49.2 | 83.0 | 57.7 | 50.7 | 71.4 | 64.6 | 52.6 | 55.1 | 34.0 | 56.2 |
| Ours | 0 | 855 | 2156% | 9.2h | 55.9 | 26.3 | 71.0 | 1321 | 51.6 | 82.5 | 63.3 | 55.9 | 74.5 | 63.1 | 47.8 | 57.3 | 35.7 | 57.8 |

Table 4: **Parametric *vs*. nonparametric visual compressor.** We follow miniGPT-4 [54] that uses Q-Former pre-trained from BLIP-2 [23] as the parametric compressor (All other aspects are maintained as in LLaVA to ensure a fair comparison). Ours: pooling with stride 64 on LLM layer 1 to ensure comparable CRs. *Our nonparametric compressor outshines the parametric Q-Former counterpart in terms of both performance and training efficiency.*

tokens). Secondly, the performance of MLLMs at the light compression level does not correlate directly with the number of visual tokens, making this observation somewhat unexpected. We attribute this to the MLLMs at this level of compression being relatively insensitive to changes in the compression ratio. This indicates that MLLMs trained at a light compression level will not hurt the model performance at all. For instance, the setting of stride 16 in light compression level attains a 188% CR and also outperforms the baseline LLaVA-v1.5-7B across all four metrics. The above observations pave the way for developing a more systematic training scheme.

| Stride | #Tokens | CR | Latency | TFLOPs | Train time | GQA | MMVet | SQA | MME | VQA$^T$ | POPE | MMB | MMB$^{CN}$ | VQA$^{v2}$ | LLaVA$^w$ | VisWiz | SEED$^I$ | MMMU | Avg. |
|---|---|---|---|---|---|---|---|---|---|---|---|---|---|---|---|---|---|---|---|
| *Heavy compression in LLM layer 2* | | | | | | | | | | | | | | | | | | | |
| 8 | 3312 | 557% | 37.9ms | 2.14 | 12.0 | 59.9$_{.13}$ | 30.1$_{.92}$ | 70.9$_{.17}$ | 1443$_{11}$ | 55.3$_{.3}$ | 85.3$_{.21}$ | 65.2$_{.25}$ | **59.5**$_{.06}$ | 76.0$_{.09}$ | 65.9$_{2.0}$ | 46.6$_{.2}$ | 62.6$_{.0}$ | 34.2$_{.54}$ | 60.3$_{.2}$ |
| 4 | 5472 | 337% | 39.1ms | 3.02 | 12.2 | 61.3$_{.23}$ | **32.3**$_{.35}$ | **71.4**$_{.16}$ | 1456$_{5.4}$ | 56.6$_{.42}$ | 85.6$_{.01}$ | 65.8$_{.54}$ | **59.5**$_{1.1}$ | 77.4$_{.02}$ | **67.3**$_{2.7}$ | 50.4$_{.38}$ | 63.9$_{.49}$ | 34.9$_{.08}$ | 61.5$_{.1}$ |
| 2 | 9792 | 188% | 48.6ms | 4.77 | 12.6 | 61.9$_{.43}$ | 30.9$_{1.1}$ | 71.6$_{.69}$ | 1450$_{18}$ | 57.6$_{.08}$ | 86.3$_{.22}$ | 67.2$_{.05}$ | 59.9$_{.4}$ | 78.0$_{.17}$ | 66.4$_{.85}$ | 48.7$_{.25}$ | 65.9$_{.49}$ | 34.1$_{.34}$ | 61.6$_{.08}$ |
| *Light compression in LLM layer 16* | | | | | | | | | | | | | | | | | | | |
| 8 | 10368 | 178% | 51.3ms | 5.00 | 12.8 | **62.6**$_{.03}$ | 30.4$_{.54}$ | 71.1$_{.27}$ | 1462$_{9}$ | 58.2$_{.01}$ | 86.0$_{.09}$ | 65.3$_{.52}$ | 58.9$_{.57}$ | 78.8$_{.12}$ | 63.9$_{1.1}$ | **51.4**$_{.15}$ | 66.8$_{.23}$ | **35.8**$_{1.4}$ | 61.8$_{.04}$ |
| 4 | 11520 | 160% | 52.2ms | 5.47 | 13.2 | 62.4$_{.10}$ | **32.0**$_{.87}$ | 70.5$_{.20}$ | 1458$_{19}$ | 58.3$_{.14}$ | 86.2$_{.15}$ | 65.9$_{.66}$ | 59.1$_{.65}$ | 78.7$_{.09}$ | 66.0$_{.57}$ | 49.6$_{1.4}$ | **67.1**$_{.09}$ | 35.0$_{.65}$ | 61.8$_{.17}$ |
| 2 | 13824 | 133% | 58.8ms | 6.40 | 14.2 | 61.9$_{.45}$ | 31.5$_{1.0}$ | 70.8$_{.49}$ | 1462$_{24}$ | **58.5**$_{.02}$ | **86.4**$_{.12}$ | **66.4**$_{.33}$ | **59.6**$_{.47}$ | 78.9$_{.02}$ | 65.3$_{.46}$ | 49.5$_{.97}$ | 66.7$_{.23}$ | 35.1$_{.87}$ | **61.8**$_{.01}$ |
| Base [27] | 18432 | 100% | 68.5ms | 8.26 | 15.3h | **62.6**$_{.49}$ | 31.9$_{1.0}$ | 70.8$_{.59}$ | **1467**$_{13}$ | 58.3$_{.15}$ | 86.1$_{.24}$ | 65.3$_{.93}$ | 59.4$_{.92}$ | **78.9**$_{.37}$ | 65.5$_{.56}$ | 49.8$_{.6}$ | 66.7$_{.25}$ | 35.1$_{.86}$ | 61.8$_{.32}$ |

Table 5: **Training MLLMs with *Visual Context Compressor* in various compression levels**. We report the average results across three runs, with the standard deviation written at the bottom right of the average result. In the heavy compression range, the performance is inversely proportional to the compression ratio. In the light compression range, the performance is not sensitive to compression. *Performance remains high for models at the light compression level.*

**Scalability to Larger Models.** As modern multimodal LLMs (MLLMs) continue to grow in size and complexity, it is crucial to determine whether the performance gains observed in smaller models can be extended to larger architectures. This ablation allows us to verify if our compression strategies maintain or even enhance their effectiveness as the model scales, ensuring their applicability to more complex real-world scenarios. As demonstrated in Tab. 6, our four-stage scheme achieved comparable performance with standard training while saving 16%(21.1 vs 17.6) training time.

| Model | #Tokens | CR | Train Time | GQA | MMVet | SQA | MME | VQA$^T$ | POPE | MMB | MMB$^{CN}$ | VQA$^{v2}$ | LLaVA$^w$ | VisWiz | SEED$^I$ | MMMU | Avg. |
|---|---|---|---|---|---|---|---|---|---|---|---|---|---|---|---|---|---|
| LLaVA-13b | 18432 | 100% | 21.1h | 63.0 | 35.0 | 74.1 | 1503 | 57.0 | 86.6 | 68.2 | 63.5 | 79.6 | 71.0 | 53.6 | 66.4 | 37.9 | 63.9 |
| Ours | 10863 | 170% | 17.6h | 63.0 | 35.4 | 74.2 | 1502 | 56.7 | 86.8 | 68.0 | 63.3 | 79.7 | 71.3 | 53.8 | 66.4 | 37.8 | 64.0 |

Table 6: **Training larger MLLMs with *LLaVolta*.** Our method achieves comparable performance across various benchmarks while reducing the training time by 16% (21.1 hours vs. 17.6 hours) and increasing the compression ratio to 170%. These results demonstrate the scalability of our approach to larger models

**Compairson with Layer-wise progressive Compression.** Given the success of stage-wise compression in accelerating training, we hypothesize that it's also beneficial for layer-wise progressive compression. To explore this, we applied nested compressors with varying strides across layers, with smaller strides in the shallower layers, where visual tokens receive more attention. As shown in Tab. 7, we experimented with a multi-stage configuration: layers 0-3 with stride=1, layers 4-11 with stride=2, layers 12-23 with stride=4, and layers 24-31 with stride=8(CR=267%). This was compared to a single-stage compression setup: layer=8, stride=8(CR=266%). While the progressive layer-wise compression showed superior performance in direct inference, it underperformed when retrained. We

attribute this to the compounded pooling of visual tokens across layers, which imposes additional challenges on the model's learning, ultimately leading to suboptimal retraining outcomes.

| Compressor | CR | GQA | MM-Vet | SQA | MME | VQA$^T$ | POPE | MMB | MMB$^{CN}$ | VQA$^{v2}$ | LLaVA$^W$ | VisWiz | SEED$^I$ | MMMU | Avg. |
|---|---|---|---|---|---|---|---|---|---|---|---|---|---|---|---|
| *Direct Inference* | | | | | | | | | | | | | | | |
| Sinlge Stage | 267% | 57.8 | 25.3 | 70.2 | 1337 | 52.1 | 86.0 | 60.4 | 52.2 | 74.6 | 56.0 | 48.1 | 58.3 | 33.3 | 57.0 |
| Multi Stage | 266% | 60.7 | 28.9 | 70.3 | 1403 | 55.4 | 85.1 | 65.2 | 57.1 | 77.7 | 60.6 | 49.1 | 64.8 | 35.2 | 60.0 |
| *Inference with Re-train* | | | | | | | | | | | | | | | |
| Single Stage | 267% | 60.7 | 30.7 | 71.3 | 1456 | 56.9 | 86.4 | 64.6 | 58.0 | 77.9 | 67.0 | 48.8 | 66.0 | 35.3 | 61.3 |
| Multi Stage | 266% | 60.9 | 29.5 | 70.5 | 1408 | 55.9 | 84.8 | 65.4 | 57.4 | 76.6 | 61.1 | 48.9 | 64.7 | 34.9 | 60.2 |

Table 7: **Comparison between single stage compressor and multi stage compressor.** mMlti-stage compression outperforming single-stage in direct inference across most tasks. However, in retrained models, multi-stage compression only shows marginal improvements, with a slight increase in the average performance.

Furthermore, we conduct an ablation study on the number of iterations in different stages (uniform *vs.* non-uniform stage splitting), which is detailed in the Appendix.

## 4.5 Extensibility to Video MLLMs

We extend our training scheme to VideoLLaVA [26] and the results in Tab. 8 reveal similar findings as before: the proposed training scheme achieve competitive results while reducing 9% training time. It is worth mentioning VideoLLaVA does not support DeepSpeed ZeRO-3, unlike LLaVA, which results in different relative efficiency gains.

| #Stages | Scheme | #Tokens$^†$ | CR$^†$ | TFLOPs$^†$ | Train-time | MSVD-QA | | MSRVTT-QA | | ActivityNet-QA | | Average | |
|---|---|---|---|---|---|---|---|---|---|---|---|---|---|
| | | | | | | Score | Acc | Score | Acc | Score | Acc | Score | Acc |
| Single | no compression | 147456 | - | 29.68 | 40.7h | 3.69 | 69.1 | 3.48 | 56.8 | 3.28 | 47.5 | 3.48 | 57.8 |
| Two | compression | 80496 | 183% | 17.73 | 37.1h | 3.71 | 69.0 | 3.50 | 56.9 | 3.29 | 47.9 | 3.50 | 57.9 |
| Three | compr. deeper | 84776 | 174% | 17.29 | 37.1h | 3.73 | 69.3 | **3.51** | **57.2** | 3.28 | 47.4 | **3.51** | 58.0 |
| Three | compr. wider | 83256 | 177% | 16.86 | 37.0h | 3.72 | 69.0 | **3.51** | **57.2** | **3.29** | 47.7 | **3.51** | 58.0 |
| Four | wider then deeper | 88704 | 166% | 18.32 | 37.2h | 3.72 | 69.1 | **3.51** | **57.2** | 3.27 | **48.0** | 3.50 | 58.1 |
| Four | deeper then wider | 86904 | 170% | 18.64 | 37.1h | **3.74** | **69.8** | 3.49 | 56.9 | 3.27 | 47.8 | 3.50 | **58.2** |

Table 8: **Performance of *LLaVolta* on VideoLLaVA[26]**. See the definition of each training scheme in Tab. 1. †: average across stages. To implement our multi-stage training, we apply the same compression processing to the 8 frames representing the video respectively. *The derived six training schemes achieve competitive results while reducing 9% training time.*

## 5 Conclusion

In this work, we conduct two initial studies to investigate and verify the redundancy of visual tokens in multi-modal LLMs. To address this, we propose *Visual Context Compressor*, a straightforward yet effective compression technique that employs a simple average pooler, seamlessly integrating into the training of MLLMs. This approach enhances training efficiency without compromising performance. To further mitigate the information loss brought by the token compression, we introduce *LLaVolta*, a multi-stage training scheme that utilizes *Visual Context Compressor* with a progressively decreasing compression rate. Experimental results on various visual question answering benchmarks verify the effectiveness of *LLaVolta* in boosting performance while demonstrating efficiency gains by reducing training costs by 16% and inference latency by 24%. To the best of our knowledge, we are the first to accelerate the training of multi-modal LLM from the compression perspective. We hope that the proposed *Visual Context Compressor* and *LLaVolta* will inspire more in-depth analysis of visual redundancy existing in current MLLMs and call for future designs of efficient training for MLLMs.

**Acknowledgement:** We thank Zhanpeng Zeng for the discussions regarding the comparison with VCC. We are also grateful for the insightful advice from our anonymous reviewers. This work was supported by a Siebel Scholarship and ONR with N00014-23-1-2641.

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

# Appendix

In the appendix, we provide additional information as listed below:

- § A provides the additional experimental results.
- § B provides the dataset information and licenses.

## A  Additional Experimental Results

### A.1  Non-uniform Stage Splitting

By default, the training time is evenly divided across each stage. To explore how the compression stage affects total training time, we modify the relative proportion of different stages. This variation is tested in the two-stage setup referenced in Tab. 1, adjusting from the standard 50% in Stage 1 and 50% in Stage 2 to different distributions. Tab. 9 below displays the results of these experiments.

| Stage 1 | Stage 2 | #Tokens | CR | GQA | MMVet | SQA | MME | $VQA^T$ | POPE | MMB | $MMB^{CN}$ |
|---------|---------|---------|------|------|-------|------|--------|--------|------|------|-----------|
| 0%      | 100%    | 18432   | -    | 62.0 | 31.1  | 70.1 | 1453.0 | 58.2   | 85.9 | 64.3 | 58.3      |
| 25%     | 75%     | 11088   | 166% | 62.1 | 31.7  | 70.6 | 1474.5 | 58.8   | 86.4 | 65.1 | 59.6      |
| 50%     | 50%     | 10863   | 170% | 62.2 | 30.0  | 70.3 | 1443.5 | 57.5   | 85.8 | 64.8 | 59.7      |
| 75%     | 25%     | 10597   | 174% | 61.6 | 32.2  | 70.8 | 1471.5 | 57.5   | 86.6 | 65.2 | 58.9      |
| 90%     | 10%     | 10407   | 177% | 61.2 | 31.0  | 70.5 | 1447.5 | 56.3   | 86.4 | 64.4 | 56.9      |
| 100%    | 0%      | 10062   | 183% | 55.9 | 29.5  | 64.1 | 1257.8 | 49.1   | 86.6 | 47.4 | 29.2      |

Table 9: **Effects of non-uniform stage splitting at the two-stage set-up**. Performance decreases as the proportion of Stage 2 decreases, albeit at the expense of lower compression ratios.

We observe that as the Stage 2 increases from 0% to 100%, there is a gradual decrease in the model's performance across various metrics (such as GQA, MMVet, SQA, MME, VQA, POPE, MMB, and $MMB^{CN}$). Although there is a decline in performance, it is relatively minor when the compression stage makes up to 50% of the training duration. However, when the proportion of the compression stage is reduced below 50%, the decline in performance becomes more significant. In conclusion, keeping the compression stage between 0-50% of the training time minimizes performance loss while still achieving significant compression ratios.

### A.2  Adaptability to Different Structures.

In addition to scaling across model sizes, it is essential to evaluate the adaptability of our approach to different model structures. As shown in Tab. 10, we conduct an experiment on Mini-Gemini [25], a structurally distinct baseline. Since Mini-Gemini employs a multi-resolution visual encoding strategy and Gemma [42] as language model. This ablation experiment assesses *LLaVolta*'s compatibility with different sophisticated visual encoding strategies.

| Model | #Tokens | CR | Train Time | GQA | MMVet | SQA | MME | $VQA^T$ | POPE | MMB | $MMB^{CN}$ | $VQA^{v2}$ | $LLaVA^w$ | VisWiz | $SEED^I$ | MMMU | Avg. |
|-------|---------|------|-------|------|-------|------|------|--------|------|------|-----------|-----------|-----------|--------|---------|------|------|
| MGM-2B | 18432 | 100% | 18.1h | 60.7 | 30.1 | 62.7 | 1327 | 57.1 | 86.0 | 61.9 | 50.6 | 76.3 | 65.9 | 48.3 | 63.8 | 28.1 | 58.3 |
| Ours | 10863 | 170% | 14.8h | 58.8 | 30.2 | 62.2 | 1325 | 54.3 | 87.0 | 62.5 | 52.5 | 76.3 | 65.7 | 48.9 | 63.1 | 27.3 | 58.1 |

Table 10: **Training struturally distinct MLLMs with *LLaVolta***. Comparison of our method with the Mini-Gemini (MGM-2B) baseline, which uses a multi-resolution visual encoding strategy. Our approach demonstrates competitive performance while reducing training time by 18% (18.1 hours vs. 14.8 hours) and achieving higher scores. This ablation highlights *LLaVolta*'s ability to adapt to different model structures and sophisticated visual encoding strategies.

# B  Datasets Information and Licenses

**GQA:**  The GQA: [20] dataset, consists of 22M questions about various day-to-day images.

License: N/A

Dataset website: `https://cs.stanford.edu/people/dorarad/gqa/download.html`

**MM-Vet:**  The MM-Vet [50] dataset, defining 6 core VL capabilities and examines the 16 integrations of interest derived from the capability combination.

License: Apache License. `https://github.com/yuweihao/MM-Vet/blob/main/LICENSE`

Dataset website: `https://github.com/yuweihao/MM-Vet/tree/main`

**SQA:**  The SQA: [31] dataset, consisting of 21k multimodal multiple choice questions with a diverse set of science topics and annotations of their answers with corresponding lectures and explanations.

License: CC BY-NC-SA (Attribution-NonCommercial-ShareAlike) `https://creativecommons.org/licenses/by-nc-sa/4.0/`

Dataset website: `https://scienceqa.github.io/#download`

**POPE**:  The POPE [24] dataset can evaluate the object hallucination in a more stable and flexible way.

License: MIT License. `https://github.com/RUCAIBox/POPE?tab=MIT-1-ov-file#readme`

Dataset website: `https://github.com/RUCAIBox/POPE`

**MMBench**:  The MMBench [30] dataset is a collection of benchmarks to evaluate the multi-modal understanding capability of large vision language models.

License: Apache License. `https://github.com/open-compass/MMBench?tab=Apache-2.0-1-ov-file#readme`

Dataset website: `https://github.com/open-compass/MMBench`

**MMBench-CN**:  The MMBench-CN [30] dataset is a collection of benchmarks to evaluate the multi-modal understanding capability of large vision language models.

License: Apache License. `https://github.com/open-compass/MMBench?tab=Apache-2.0-1-ov-file#readme`

Dataset website: `https://github.com/open-compass/MMBench`

**MME**:  The MME [13] dataset containing 30 images with 60 instruction-answer pairs for coarse-grained recognition task; 917 images for fine-grained recognition task; 20 images with 40 instruction-answer pairs for OCR task.

Dataset website: `https://github.com/QwenLM/Qwen-VL/blob/master/eval_mm/mme/EVAL_MME.md`

License: Tongyi Qianwen LICENSE AGREEMENT. `https://github.com/QwenLM/Qwen-VL/tree/master?tab=License-1-ov-file#readme`

**TextVQA**:  The TextVQA [39] dataset containing 30 images with 60 instruction-answer pairs for coarse-grained recognition task; 917 images for fine-grained recognition task; 20 images with 40 instruction-answer pairs for OCR task.

Dataset website: `https://github.com/facebookresearch/mmf.git`

License: BSD LICENSE. `https://github.com/facebookresearch/mmf/blob/main/LICENSE`

**VQA-v2:**  The VQA-v2 [14] dataset, containing 265,016 images, dataset containing open-ended questions about images. These questions require an understanding of vision, language and common-sense knowledge to answer.

License: Commons Attribution 4.0 International License. `https://visualqa.org/terms.html`

Dataset website: `https://visualqa.org/`

