# OpenReview forum: "Efficient Large Multi-modal Models via Visual Context Compression"
_NeurIPS.cc/2024/Conference — NeurIPS 2024 poster_

### Official Review · Reviewer_PsS7 · 2024-06-30

**Soundness:** 3
**Presentation:** 2
**Contribution:** 3
**Rating:** 5
**Confidence:** 4

**Summary:**

This paper shows that visual tokens are redundant in MLLM and can be compreseed by a large ratio without significantly hurting the model performance. Based on this observation, this paper studied several different approaches to compress visual tokens and identified that the simple average pooling method is the most effective one. Based on this, the paper further studied several different stage-wise MLLM training strategies, where the training starts with heavy compression and ends with no compression. The proposed training strategy could save the training cost by 16% while achieving even better performance than the baseline.

**Strengths:**

1. The observation that visual tokens are redundant and can be largely compressed without significantly hurting the model performance is good. It could serve as a direction for future works.

2. The paper conducted thorough empirical studies about different compression methods, and proposed several training methods based on visual token compression.

3. The proposed method shows better performance than the baseline while reducing the training cost by 16%.

**Weaknesses:**

1. It would be better to provide more discussions and investigations on why other visual compressors are significantly worse than average pooling. This may provide more insights on how to design good compressor.

2. The paper only studied LLaVA-1.5-7B. It would be better to show that if the method could scale up to larger models like 13B, 34B, or other structures such as Mini-Gemini [1]. Showing the efficiency on those large variants can better demonstrates the effectiveness of the method.

3. The 16% training efficiency improvement is marginal in practice, especially given that the four-stage training would be more cumbersome compared to the baseline.

[1] Li, Yanwei, et al. "Mini-gemini: Mining the potential of multi-modality vision language models." arXiv preprint arXiv:2403.18814 (2024).

**Questions:**

Please see the weakness section

**Limitations:**

Yes

---

> ### Author Rebuttal · Authors · 2024-08-07
>
> We appreciate the reviewer's constructive suggestions. We address the concerns raised on a point-by-point basis, including additional benchmarks on the global response PDF. We will include the new results in our revised manuscript.
>
> For weakness 1: "more discussions and investigations on why other visual compressors are significantly worse than average pooling", we thanks for the reviewer's suggestion.  In line 262-263, we analyzed that "they are ineffective when applied to training because the in-training attention scores are unstable.". Our insight is that while advanced compressors (e.g., attention-based token pruning) excel in inference-only scenarios (See bottom rows in Table 2), the simple pooling method performs better during training (See top rows in Table 2). We hypothesize that this is because training advanced compressors, such as attention-based pruning, necessitates (1) differentiable token selection and (2) stable attention mechanisms of LM Transformers.
>
>
> For weakness 2:  "scale up to larger models like 13B, 34B, or other structures such as Mini-Gemini [1]", we thanks for the reviewer's suggestion.  To show the scalability, we scale the method up to 13B model, and observe consistent performance and efficiency improvements.
> | Scheme    | #Token | CR | train-time  | GQA  | MM-Vet | SQA  | MME  | VQA^T | POPE | MMBench | MMB-CN | VQAv2 | LLaVA^W | VisWiz | SEED^I | MMMU | Avg. |
> |---------------|---------|------|---|---|--------|------|------|-----------|------|------|-----------|------------|-------------|--------|---------|------|------|
> | LLaVA-13B | 18432             | 100%            | 21.1h      | 63.0 | 35.0  | 74.1 | 1503 | 57.0      | 86.6 | 68.2 | 63.5       | 79.6       | 71.0        | 53.6   | 66.4       | 37.9 | 63.9  |
> | Ours-13B  | 10863             | 170%         | 17.6h      | 63.0 | 35.4  | 74.2 | 1502  | 56.7      | 86.8 | 68.0 | 63.3       | 79.7       | 71.3        | 53.8   | 66.4       | 37.8 | 64.0  |
>
>
> To show the generalizability to different structures, we supplement the experiment on Mini-Gemini (MGM-2B). Our four-stage compression training delivers comparable results with a 17% increase in training efficiency and significantly fewer vision tokens as input. We plan to incorporate additional structures in future versions.
>
> | Scheme    | #Token | CR   | GQA  | MM-Vet | SQA  | MME  | VQA^T | POPE | MMBench | MMB-CN | VQAv2 | LLaVA^W | VisWiz | SEED^I | MMMU | Avg. |
> |---------------|---------|------|------|--------|------|------|-----------|------|------|-----------|------------|-------------|--------|---------|------|------|
> | MGM-2B      | 18432    | 100% | 60.7 | 30.1 | 62.7 | 1327  | 57.1 | 86.0 | 61.9 | 50.6 | 76.3 | 65.9 | 48.3 | 63.8 | 28.1 | 58.3 |
> | Ours      | 10863    | 170% | 58.8 | 30.2 | 62.2 | 1325 | 54.3 | 87.0 | 62.5 | 52.5 | 76.3 | 65.7 | 48.9 | 63.1 | 27.3 | 58.1 |
>
>
> We thanks for the reviewer's comments on Weakness 3. It's worth noting that our method not only improves training efficiency by 16% but also enhances average performance by 0.5\% across 13 benchmarks. To address the cumbersomeness, we are exploring the development of a simple training schedule (linearly decreasing the compression ratio over time; like learning rate schedule), aimed at balancing training efficiency and model simplicity.
>
> We sincerely hope our rebuttal addresses your concerns, and we look forward to your feedback. Your response will motivate us to refine our paper into a more solid version.
>
> Kind regards,
>
> Authors of Paper 2615

---

### Official Review · Reviewer_ENde · 2024-07-14

**Soundness:** 4
**Presentation:** 4
**Contribution:** 3
**Rating:** 6
**Confidence:** 4

**Summary:**

The paper presents a novel approach to reducing redundancy in visual tokens within MLLMs by introducing the Visual Context Compressor (VCC) and Stage-wise MLLM Training. The VCC uses simple average pooling to compress visual tokens during training, enhancing efficiency without sacrificing performance. The Stage-wise MLLM Training progressively reduces compression as training proceeds, ensuring no loss of information during testing. The proposed methods demonstrate significant improvements in training efficiency and performance across multiple benchmarks.

**Strengths:**

1. The paper addresses an underexplored area in MLLMs by focusing on the redundancy of visual tokens and introducing effective compression techniques.
2. The proposed methods achieve significant reductions in training costs while maintaining or improving performance on various benchmarks.
3. The paper provides thorough experimental validation across multiple benchmarks, demonstrating the effectiveness of the proposed methods.

**Weaknesses:**

1. While the paper shows improvements across several benchmarks, the evaluation is primarily focused on visual question answering tasks. Additional evaluation on other multi-modal tasks could strengthen the claims.
2. The paper mentions potential information loss due to compression but does not provide a detailed analysis/visualization of how this might affect tasks requiring dense visual information.
3. This paper lacks some essential experiments. (1) The paper does not explore the effect of using larger input resolutions. For example, evaluating the method with 448×448 input images, which contain 2.25 times more visual tokens than the 336×336 input, could provide insights into the method's scalability. Designing a compression setting with a ratio of 225% for this input size and comparing it with the original LLaVA setting (compression ratio of 100%) would be valuable. (2) Testing the method on larger models such as LLaVA-13B or models with more input visual tokens (e.g., LLaVA-Next) could solidify the experimental section and demonstrate the robustness of the proposed approach.
4. The best stage-wise MLLM training scheme looks difficult to transfer to other model & data settings. The training scheme has too many options and variables. If the model size and dataset size increase significantly, the time cost and computation cost for finding the best scheme can become prohibitively large.

**Questions:**

Please see the weakness.

Minor issues:

1. What does the #Tokens mean in the tables?  Is it inversely proportional to CR?
2. How do you compute the average performance in Table 5?
3. Some suggestions for future improvements: (1) multi-stage compression: use different compressors’ strides in the different positions of the LLM. For example, layer 1-3: stride=1; layer 4-12: stride=2; layer 13-24: stride: 4. (2) extend this technique to other modalities, such as video understanding.

**Limitations:**

The authors have adequately discussed the limitations.

---

> ### Author Rebuttal · Authors · 2024-08-07
>
> We appreciate the reviewer's constructive suggestions. We address the concerns raised on a point-by-point basis, including additional benchmarks on the global response PDF. We will include the new results in our revised manuscript.
>
> To address weakness 1, we follow the reviewer's suggestion, and add the evaluation of multi-modal tasks in Table 1 of Global Response, especially MMBench and MMMU (A Massive Multi-discipline Multimodal Understanding and Reasoning Benchmark for Expert AGI)[49].
>
> For weakness 2, we thanks for the reviewer's comment on information loss. We have discussed this in the limitation section of original paper. The naive compression may not be ideal for tasks requiring dense comprehension, grounded reasoning, and OCR/text capabilities. Nonetheless, our staged training approach, which does not involve compression in the final stage, is capable of managing tasks that require dense visual information. We provide analysis and visualization in Global Response.
>
> To address weakness 3.1, we try to implement the experiment of larger 448×448 input resolution. However, we note that the default openai-CLIP-336 encoder can only input 336x336 and interpolating positional embedding to 448x448 degrades the performance, resulting in -9% drop in GQA. We are experimenting the dynamic high resolution of LLaVA-NeXT (its training code has not yet been made public as of Aug/8) by employing openai-CLIP-224 to encode four 224x224 images derived from a 448x448 image. We plan to include the large input experiment in the revision once our new results are available.
>
> For weakness 3.2, we thanks for the reviewer's suggestion.  As LLaVA-NeXT does not make the training code public upon the submission of rebuttal, we are in the progress of reproducing the code. We supplement the experiment of testing the method on larger model of LLaVA-13B in Table 3 of Global Response, and we can achieve 177% compression rate and accelerate training efficiency by 15% while keeping the performance. We will include the new results in our revision.
>
> For weakness 4, we thanks for the reviewer's suggestion. We aimed at providing comprehensive analysis of design space. We will provide a simple version for the practical use in the near future. For instance, we might implement a universal training schedule that gradually reduces the compression ratio over time.
>
> We thanks for the reviewer's comments for the minor issues. We address them as follows.
>
> Q1. We thanks for the reviewer's comment. #Tokens means the summation of number of visual tokens in each LLM transformer layer. It is inversely proportional to CR.
>
> Q2.  We mentioned this in line 242: "When reporting average performance in Tab. 5, the score of MME is normalized by 2000, as its range is from 800 to 2000". The average performance is simply the mean of the (normalized when it is MME) individual performances.
>
> Q3 (1) multi-stage compression. We thanks for the reviewer's suggestion. The multi-stage compression is a great idea. We follow the suggestion and experiment with the setting of multi-stage (layer 0-3: stride=1; layer 4-11: stride=2; layer 12-23: stride: 4; layer 24-31: stride: 8), resulting in a CR of 267%. We compare this to single-stage compression (on layer 8 stride 8) with a CR of 266%.
>
> | Compressor    | Phase | CR   | GQA  | MM-Vet | SQA  | MME  | VQA^T | POPE | MMBench | MMB-CN | VQAv2 | LLaVA^W | VisWiz | SEED^I | MMMU | Avg. |
> |---------------|---------|------|------|--------|------|------|-----------|------|------|-----------|------------|-------------|--------|---------|------|------|
> | layer=8 stride=8 | inference | 266%| 57.8|25.3|70.2|1337|52.1|86.0|60.4|52.2|74.6|56.0|48.1|58.3|33.3|57.0|
> | layer=8 stride=8| training| 266%| 60.7 | 30.7 | 71.3 | 1456 | 56.9 | 86.4 | 64.6 | 58.0 | 77.9 | 67.0 | 48.8 | 66.0 | 35.3 | 61.3 |
> | multi-stage | inference| 267% | 60.7 | 28.9 |70.3 | 1403 | 55.4 | 85.1 | 65.2 | 57.1 | 77.7 | 60.6 | 49.1 | 64.8 | 35.2 | 60.0 |
> | multi-stage| training| 267% | 60.9 | 29.5 |70.5 | 1408 | 55.9 | 84.8 | 65.4 | 57.4 | 76.6 | 61.1 | 48.9 | 64.7 | 34.9 | 60.2 |
>
> The multi-stage compressor exhibits strong performance when applied directly during inference, outperforming layer 8 stride 8 by 3% under same CR. However, it's surprising that training the multi-stage compressor yields only a marginal average performance improvement of 0.2%. We analyze that the complexity of multi-stage operations makes the LLM more challenging to train. We will have more future analysis.
>
> Q3 (2) video extension. We thanks for the reviewer's suggestion.  For extension of modalities, we have experimented on video-language understanding (based on video-LLaVA). We observe consistent enhancements (an average improvement of +0.4\% across three video datasets and a 9% reduction in training time) over the baseline using our new training setting.  We will include the new results in our revision.
>
>
> | Scheme           | #Tokens | CR | TFLOPs | Train-time | MSVD-QA Score | MSVD-QA Acc | MSRVTT-QA Score | MSRVTT-QA Acc | ActivityNet-QA Score | ActivityNet-QA Acc | Average Score | Average Acc |
> |------------------|-------------------|--------------|------------------|------------|---------------|-------------|-----------------|---------------|----------------------|--------------------|---------------|-------------|
> | Video-LLaVA-7B   | 147456      | 100%  | 29.68   | 40.7h      | 3.69     | 69.1        | 3.48            | 56.8          | 3.28                 | 47.5               | 3.48          | 57.8        |
> | Ours             | **86904**  |  **170%** | **18.64**            | **37.1h**      | 3.74      | 69.8    | 3.49           | 56.9          | 3.27                 | 47.8               | **3.50**          | **58.2**
>
>
> We sincerely hope our rebuttal addresses your concerns, and we look forward to your feedback. Your response will motivate us to refine our paper into a more solid version.
>
> Kind regards,
>
> Authors of Paper 2615

---

### Official Review · Reviewer_dJy4 · 2024-07-15

**Soundness:** 3
**Presentation:** 3
**Contribution:** 3
**Rating:** 5
**Confidence:** 4

**Summary:**

The paper presents a compelling study on redundancy of visual tokens in MLLMs and practical approaches to reduce them. The paper first verifies that one could eliminate up to 70% of visual tokens at testing time via simple average pooling with minial performance degradation. Then they experiments with several approaches for compressing the visual tokens, finding that a simple average pooling works the best. They also propose a staged training recipe, where we could save computation during early training stages and gradually remove compression.

**Strengths:**

1. The idea is clean and simple. It is nice to empirically verify the redundancy in visual tokens.
2. The proposed staged training and the discussion around wider then deeper v.s. deeper then wider is interesting.

**Weaknesses:**

- There seems to be an easy baseline missing from the discussion, such as 2D conv (as in Honeybee).
Honeybee: Locality-enhanced Projector for Multimodal LLM
- There are some ambiguous experimental details. Please see questions.

**Questions:**

1. When doing average pooling, why choose the 1D average pooling instead of 2D average pooling over the grid features?
2. What is K in table 2? I cannot seem to be able to find discussion on tuning k v.s. stride in Section 4.2.
3. For ablation, why report only the performance on the 4 benchmarks? I suspect that the compression performance will also be quite dataset-dependent.

**Limitations:**

Yes.

---

> ### Author Rebuttal · Authors · 2024-08-07
>
> We appreciate the reviewer's constructive suggestions. We address the concerns raised on a point-by-point basis, including additional benchmarks on the global response PDF. We will include the new results in our revised manuscript.
>
> For weakness 1, we have included 2D-Conv and C-Abstractor (from Honeybee) as a baseline in Table below. The stride of 2D-Conv and C-Abstractor are the same with other compressor to ensure fair comparison under same compression ratio (CR). However, we found that (1) The performance of convolution-based compressors is significantly lower, with 2D-Conv achieving 55.1\%, C-Abstractor achieving 50.6\%, and 1D Pool reaching 60.4\% in average accuracy across 13 benchmarks. (2) there is overhead parameters of conv. We analyze that it is hard to optimize the extra convolution kernel within LLM at the same time (this is an underexplore area).
>
> | Compressor    | #Tokens | CR   | GQA  | MM-Vet | SQA  | MME  | VQA^T | POPE | MMBench | MMB-CN | VQAv2 | LLaVA^W | VisWiz | SEED^I | MMMU | Avg. |
> |---------------|---------|------|------|--------|------|------|-----------|------|------|-----------|------------|-------------|--------|---------|------|------|
> | 2D-Conv       | 2232    | 826% | 58.6 | 28.6   | 71.6 | 1366 | 51.8      | 84.4 | 63.8 | 55.6      | 74.0       | 63.8        | 48.1   | 60.1    | 25.6 | 55.1 |
> | C-Abstractor  | 2232    | 826% | 53.7 | 23.7   | 70.9 | 1209 | 48.7      | 82.8 | 58.0 | 50.3      | 68.2       | 48.6        | 48.0   | 53.4    | 23.9 | 50.6 |
> | 2D Pool       | 2232    | 826% | 57.5 | 28.7   | 71.5 | 1426 | 53.1      | 84.0 | 64.2 | 58.7      | 74.3       | 64.2        | 50.0   | 66.5    | 34.3 | 56.1 |
> | 1D Pool       | 2232    | 826% | 58.3 | 29.2   | 71.4 | 1434 | 53.6      | 83.8 | 64.8 | 58.6      | 74.5       | 65.0        | 49.1   | 66.8    | 35.0 | 56.3 |
>
> It is worthy mentioning that Honeybee focus on the design of projector outside LLM while our focus is the visual context compression within LLM. Therefore, our method has the potential to accelerate the inference and training of Honeybee (i.e., keep Honeybee's 2D Conv on projector before LLM, and add our 1D pool compressor on visual tokens within LLM). We will include these new results in our revision.
>
> For question 1, we thank the reviewer for the comments. We add experiment with 2D pooling in the same Table above. As results, 1D pooling perform slightly better than 2D pooling (+0.2\%) under the same compression ratio.
> This could be due to the visual tokens being processed in a 1D manner within several LLM layers. Besides, we opted for 1D pooling because it allows for more adaptable compression ratio. For instance, using a 1D pool with a stride can reduce the number of tokens by 2x, whereas a 2D pool with a stride of 2 reduces the token count by 4x. We will include these new results in our revision.
>
> For question 2, we appreciate the reviewer's feedback. K is set to 2 with stride set to 8 in Table 2, resulting in a compression ratio of 556\% to approximate [41]’s 514\% (fixed CR) for a fair comparison of compressor.
>
> For question 3,  we thank the reviewer for the comments. The previous validated four benchmarks are considered representative because they assess diverse capabilities: GQA evaluates visual understanding, SQA focuses on scientific knowledge across 26 topics, MME tests perception and cognition through 14 subtasks, and MM-Vet measures ensembling skills in recognition, OCR, math, and spatial awareness (encompassing other benchmarks like VQAv2, COCO, and TextVQA).
>
> In Table 1 of GLOBAL RESPONSE PDF, we have included all 13 benchmarks in our ablation now and evaluated based on the average performance across these benchmarks.  The evaluated 13 benchmarks include not only VQA but also multimodal benchmarks. We achieve consistent conclusions with our original paper for both the ablation study and final results. We will include these new results in our revision.
>
> We sincerely hope our rebuttal addresses your concerns, and we look forward to your feedback. Your response will motivate us to refine our paper into a more solid version.
>
> Kind regards,
>
> Authors of Paper 2615

---

### Author Rebuttal · Authors · 2024-08-07

We are grateful to the reviewers for their feedback. Reviewer dJy4 commends the paper for presenting "The paper presents a compelling study. The idea is clean and simple". Reviewer ENde notes "the paper addresses an underexplored area in MLLMs", and Reviewer PsS7 appreciates the thorough empirical studies conducted. Additionally, Reviewer PsS7 suggests that the work could provide a direction for future research, highlighting its potential to influence further advancements.

We thanks the reviewers for the constructive comments. To address them, we provide **a global response PDF (attached to this post)** with more benchmark results (dJy4, ENde), scalability to large 13B model (ENde, PsS7), and additional visual analysis (ENde). We also address each reviewer's questions with new results on a point-by-point basis.

We sincerely hope our rebuttal addresses your concerns, and we look forward to your feedback. Your response will motivate us to refine our paper into a more solid version.

Kind regards,

Authors of Paper 2615

---

### Author Response · Authors · 2024-08-12
**Invitation for Discussions and Clarifications**

Dear Reviewers,

We wholeheartedly invite and encourage discussion. To ensure we have adequately addressed your concerns, could you please review our responses and provide any further comments or questions you may have? We are more than willing to address any additional concerns.  Your time and expertise are sincerely appreciated, and we eagerly await your continued input.

Kind regards,

Authors of Paper 2615

---

> ### Comment · Area_Chair_MiZt · 2024-08-13
>
> Dear reviewers,
>
> Please carefully read the rebuttal and the other reviews, then reply to the authors indicating whether your questions/concerns have been addressed. If not, please specify which questions/concerns remain unresolved so that the authors have a fair opportunity to address them before the author-reviewer discussion period ends in approximately 36 hours from now (August 13th, 11:59pm AoE).
>
> Best,
> AC

---

> > ### Author Response · Authors · 2024-08-13
> > **Invitation for Discussions and Clarifications**
> >
> > Dear reviewers,
> >
> > We wholeheartedly invite and encourage discussion. We would be grateful for any further questions or comments you might have, and are fully prepared to provide additional details or clarifications. Your feedback is invaluable to enhancing the quality of our work.
> >
> > As AC mentioned, we are dedicated to maximizing the remaining time in the author-reviewer discussion period to address any additional concerns you may have.
> >
> > Authors of paper 2615

---

### Decision · Program_Chairs · 2024-09-25

**Decision:**

Accept (poster)

**Comment:**

This paper presents a study on the analysis of redundancy concerning visual tokens and efficient training within large multimodal models. The paper received 2 borderline accepts and 1 weak accept recommendations from reviewers. Positive points included a clean and simple idea, thorough empirical studies on an underexplored area, and significant reductions in training costs while maintaining or improving performance. Negative points included lacking discussions and investigations on comparisons to other visual compressors, missing experiments for more benchmarks, larger input resolutions, and larger models, and marginal efficiency improvements.  There were also questions including the use of 1D average pooling instead of 2D average pooling, and other questions regarding some approach and experiment details.  Although the reviewers did not follow up post-rebuttal, the ACs feel that most of the major concerns and questions were adequately addressed by the rebuttal. Overall, after carefully considering the paper and rebuttal, the ACs feel that the paper's strengths outweigh the negatives, and thus recommend accept. Please incorporate the rebuttal points into the camera ready version.